# Response process validity of three patient reported outcome measures for people requiring kidney care: a think-aloud study using the EQ-5D-5L, ICECAP-A and ICECAP-O

Paul Mark Mitchell  ,[1] Fergus John Caskey,[2,3,4] Jemima Scott,[3,4] Sabina Sanghera,[1] Joanna Coast[1]

[1]Health Economics Bristol, Population Health Sciences, University of Bristol, Bristol, UK
[2]Population Health Sciences, University of Bristol, Bristol, UK
[3]UK Renal Registry, Southmead Hospital Bristol, Bristol, UK
[4]North Bristol NHS Trust, Southmead Hospital, Bristol, UK

**Correspondence to**
Dr Paul Mark Mitchell;
paul.mitchell@bristol.ac.uk

## ABSTRACT

**Objectives** To determine the response process validity, feasibility of completion, acceptability and preferences for three patient-reported outcome measures that could be used in economic evaluation—the EQ-5D-5L, ICECAP-A and ICECAP-O—in people requiring kidney care.

**Design** Participants were asked to 'think-aloud' while completing the EQ-5D-5L, ICECAP-A and ICECAP-O, followed by a semistructured interview. Five raters identified errors or struggles in completing the measures from the think-aloud component of the transcripts. Patient preferences for measures were extracted from the semistructured interview.

**Setting** Eligible patients were identified through a large UK secondary care renal centre.

**Participants** In total, 30 participants were included in the study, consisting of patients attending renal outpatients for chronic kidney disease (n=18), with a functioning kidney transplant (n=6) and receiving haemodialysis (n=6).

**Results** Participants had few errors and struggles in completing the EQ-5D-5L (11% error rate, 3% struggle rate), ICECAP-A (2% error rate, 2% struggle rate) and ICECAP-O (4% error rate, 3% struggle rate). The main errors with the EQ-5D-5L were judgements that did not comply with the 'your health today' instruction. Comprehension errors were most prominent on ICECAP-O. Judgement errors were the only errors reported on ICECAP-A. Although the EQ-5D-5L had slightly more errors and struggles, it was the measure most preferred, with participants able to make a clearer link with EQ-5D-5L and their health condition.

**Conclusions** The EQ-5D-5L, ICECAP-A and ICECAP-O are feasible for people requiring kidney care to complete and can be included in studies conducting economic evaluations of kidney care interventions. Further research is required to assess how health (eg, EQ-5D) and capability (eg, ICECAP) measures can be included in an economic evaluation simultaneously, as well as what ICECAP measure(s) to include when patient groups straddle the age ranges for ICECAP-A (18 years and older) and ICECAP-O (65 years and older).

## Strengths and limitations of this study

► This is the first study to include EQ-5D-5L, ICECAP-A and ICECAP-O in a think-aloud study with people requiring kidney care.
► The sample consists of a broad range of people requiring kidney care including renal outpatients for chronic kidney disease, kidney transplant check-ups and haemodialysis.
► Think-aloud studies aim to identify errors and struggles in task completion as they occur.
► Five raters, with diverse experience across health economics, qualitative research and kidney care, identified errors and struggles from the think-aloud transcripts.
► Think-aloud relies on participants verbalising their difficulty in task completion.

## BACKGROUND

Healthcare expenditure is rising globally and has been increasing at a faster rate than international economic growth over the past decade.[1] Chronic kidney disease (CKD) is a growing burden on healthcare resources. In the 2015 Global Burden of Disease Study,[2] CKD were the 12th leading cause of death and 17th leading cause of global life years lost.[3] In the UK alone, CKD accounts for more than one per cent of the National Health Service (NHS) annual budget.[4] Given this volume of expenditure, it is important that any healthcare resources allocated to managing kidney problems are used efficiently.

To determine which interventions should be recommended for practice, economic evaluations provide evidence on cost-effectiveness by comparing the costs and benefits of alternative interventions. In health and care, these economic evaluations increasingly rely on patient-reported outcome

measures (PROMs) to capture the health-related quality of life improvements from interventions[5] and are recommended for the generation of quality adjusted life years (QALYs) internationally.[6] A QALY is a combination of life years adjusted for health-related quality of life.[7] Choice of PROM in generating QALYs plays an influential role in deciding if a treatment is cost-effective.[8 9] The EQ-5D is the most widely used measure to calculate QALYs in economic evaluations internationally[5] and has been translated into 169 different languages.[10] The EQ-5D has also been separately recommended by an expert consensus for routine collection across European renal registries.[11]

Despite international recommended use of QALYs in healthcare, the suitability of this outcome is debated, partly due to the exclusive focus on health gains and not broader well-being.[7 12 13] An alternative approach has been proposed to capture broader well-being, which focuses on a person's capabilities, meaning a person's freedom to achieve the things in life that are valuable to them.[14] Health bodies in the UK and the Netherlands have recognised the limitation of relying purely on QALYs in social care[15] and long-term health conditions.[16] Capability measures, such as the ICECAP-A[17] (A— all adults aged 18 years and above) and the ICECAP-O (O—older adults aged 65 years and above),[18] have been recommended as ways to capture the broader benefits for these patient groups. It is not entirely clear, however, which ICECAP measure to use when the age range of a patient group could use either ICECAP-A or ICECAP-O. People requiring kidney care are a prime example of this challenge, with the median age for starting renal replacement therapy in the UK being 64 years of age in 2017.[19] A recent study found the ICECAP-O to be a valid measure in over 75-year-old patients receiving dialysis or conservative care for end-stage kidney disease (ESKD).[20] ICECAP-O was also developed first and has been shown to be a valid outcome in older and younger adults in different settings.[21] However, no previous study has tested the ICECAP-A and ICECAP-O in the same patient group.[21]

The objective of this study was to (1) assess response process validity, feasibility of completion and acceptability of the health-related quality of life PROM EQ-5D-5L, and two capability PROMs of broader well-being, ICECAP-A and ICECAP-O, in patients requiring kidney care, and (2) assess patient preferences for the three PROMs.

## METHODS
This research consists of a 'think-aloud' study followed by a semistructured interview. A think-aloud study is a cognitive interview method whereby individuals are asked to verbalise their thought process when completing measures.[22] Think-aloud interviews enable the examination of problems patients may encounter in terms of comprehension, retrieval, judgement and response difficulties. The interviewer remains silent, so long as individuals continue to think-aloud. This process is thought to give a more realistic picture of the problems that individuals face when completing questionnaires than more direct interview methods that interrupt task completion.[23] Think-aloud interviews are a method that allow for the assessment of validity in terms of investigating response processes.[24] Assessing response processes is one of five recommended sources of validity evidence.[25]

### Sampling and recruitment
Samples for previous think-aloud studies on health and capability PROMs have ranged from 10[26] to 34[27] participants. Based on these previous studies, saturation (whereby no new insights would be anticipated from additional sampling)[28] was expected to be reached at 25 participants here.

Patients were recruited through a large UK secondary care renal centre. Participants were sampled purposefully to achieve diversity in age (classified as <65 or ≥65) and type of kidney care received, but in line with general approaches to sampling in qualitative research, sampling did not aim for representativeness.[28] Sampling was conducted through renal outpatient lists and a dialysis unit. Eligibility required individuals to have CKD, be willing and able to provide informed consent to participate, and be able to communicate in English (because the study was exploring the use of English language questionnaires). Potential participants received a participant information sheet (PIS— see online supplementary file 2) in the post or at the dialysis unit and were invited to take part via a follow-up telephone call from a clinical trials officer. The PIS was the only information provided to the participant about the researcher prior to interview.

### Instruments investigated
The EQ-5D-5L consists of five dimensions of health status covering mobility, self-care, usual activities, pain/discomfort and anxiety/depression across five levels ranging from no problems to extreme problems.[29] The EQ-5D-5L was introduced to supersede the EQ-5D-3L to reduce ceiling effects and increase sensitivity to change, by moving from a three-level to a five-level severity measure of health problems, but with the same five dimensions. A Visual Analogue Scale (VAS) is also included that asks respondents to rate their health today on a 0–100 scale from worst to best imaginable health state. The National Institute for Health and Care Excellence in England recommends the collection of the EQ-5D-5L for conducting health economic evaluation.[30]

The ICECAP-A is a capability well-being measure developed for the general adult population (ie, all adults, including those aged over 65).[17] It consists of five dimensions relating to a person's capability to have attachment, stability, achievement, enjoyment and autonomy. Each dimension has four levels ranging from no capability to full capability. The capabilities were identified through qualitative research with members of the general public aged 18 years and above (including over 65 year olds) to identify what was most important to them in their life.[17]

The ICECAP-O is a capability well-being measure developed for older adults.[18] The ICECAP-O was the first of the ICECAP suite of measures developed that aimed to develop a more appropriate quality of life measure for older adults specifically for use in the economic evaluation of health and care interventions.[31] It consists of five dimensions relating to a person's capability to have attachment, security, role, enjoyment and control. Each dimension has four levels ranging from no capability to full capability. As with the ICECAP-A, the capabilities were identified through qualitative research, but in this case with older members of the general public aged 65 years and above.[32]

## Data collection

Once participants had provided informed consent, interviews took place at the renal centre or in the participant's home. All interviews were conducted by PM, a male PhD researcher in health economics with qualitative interview training and an interest in PROMs research. Initial questions focused on basic sociodemographic information. Participants then completed a simple warm-up task to determine the number of windows in their home. A second warm up task involved the completion of the Global Quality of Life scale.[33] Participants then completed the think-aloud exercise. They were allocated sequentially to receive ICECAP-A or ICECAP-O first or third, with EQ-5D-5L (including the EQ-VAS) always completed second; given the similarities between ICECAP-A and ICECAP-O, it was seen as a stronger design to separate these two measures to avoid confusion. Participants were not interrupted during the completion of the three measures unless they were silent for longer than 10 seconds when they were asked to 'keep thinking aloud'. Following the think-aloud task, a semistructured interview was conducted to clarify issues arising in the think-aloud task and to explore views about the three measures. Field notes were made during the think-aloud component to guide the semistructured interview. The interview guide was piloted prior to interview. Transcripts were not returned to participants for comment and/or correction and they did not provide feedback on the study findings. Repeat interviews were not carried out. Interviews were audio recorded and transcribed verbatim. Data were managed in Microsoft Word and Excel.

## Data analysis
### Think-aloud analysis

The think-aloud section from each transcript was extracted for each of the measures and divided into 16 segments: 6 representing the items on the EQ-5D-5L (including the EQ-5D-VAS), 5 items on the ICECAP-A and ICECAP-O, respectively. Think-aloud sections of the interview, alongside the reported response level for each item on each measure, were presented to five independent raters (PM, FC, JS, SS and JC), with expertise in health economics (PM, SS and JC), qualitative research (JC) and renal care (FC and JS). Each rater individually examined all think-aloud

sections to identify problems participants encountered when completing each of the three measures. Raters were asked to identify whether responses were error free or contained any of the following problems, based on the survey response model[34]:

1. Comprehension error (understanding the question in the way the researcher intended).
2. Retrieval error (retrieving appropriate information from their long-term memory).
3. Judgement error (correctly judging how recalled information should be used to answer).
4. Response error (format the information into a valid response for the questionnaire).
5. Struggle (not one of the four errors but clear difficulty in answering the question).[27]

Following these independent ratings, each item was identified as error free, containing an error or containing a struggle, using the following rules:

▸ Where three or more raters identified a specific error/struggle, it was classed as an error/struggle.
▸ Where one or none thought an error was present, it was marked as error free.
▸ Where two or more raters identified an error/struggle but there was no majority agreement on the type of error/struggle, a decision was made during a consensus meeting with all raters; a majority decision was used when no consensus occurred.

Consistency between raters on the coding of the data was assessed using raw agreement and a weighted kappa statistic.[35] For the latter, where an error and no error were reported between raters, this was weighted as 0; all other disagreements—such as different error types, error/struggle or struggle/no error—was weighted as 0.5, with agreement weighted as 1.

### Preference between measures

During the semistructured interviews following the think-aloud task, individual preferences for completing the three measures were explored. Individuals were asked which of the three measures they preferred and why they thought it was more important in assessing their quality of life.

### Patient and public involvement

Patients and the public were not directly involved in the design of the study.

## RESULTS

Three hundred and thirty-four patients were invited to take part in the study. Of these, 161 responded to telephone follow-up and 37 agreed to participate. In four cases, patients did not attend the interview, one individual was too unwell to participate and one decided against participation during the consent process. Thirty-one individuals took part, but one individual did not understand the task (reading aloud their response levels only), leaving 30 individuals as the final sample. Most

**Table 1** Participants' characteristics (n=30)

| | |
|---|---|
| **Sex** | |
| Male | 23 |
| Female | 7 |
| **Ethnicity** | |
| White | 28 |
| Non-white | 2 |
| **Age group** | |
| 75+ | 4 |
| 65–74 | 8 |
| 55–64 | 7 |
| 45–54 | 6 |
| 35–44 | 4 |
| 18–34 | 1 |
| **Kidney care received** | |
| Renal outpatients | 18 |
| Renal outpatients (transplant) | 6 |
| Dialysis | 6 |

interviews took place at the healthcare facility, with four taking place in the participant's home. Most interviews were conducted one-to-one; on occasion at the healthcare facility patients' partners were present. Interviews were conducted between April and July 2017 and lasted between 16 and 55 min (average 33 min). Characteristics of the sample are presented in table 1.

### Think-aloud analysis: errors and struggles
Following independent coding of the think-aloud interviews by five raters, inter-rater agreement was similar for ICECAP-A (85%–95%) and ICECAP-O (83%–93%), slightly lower for EQ-5D-5L (78%–84%) and weighted chance-corrected agreement being rated 'fair' to 'moderate' for 29 out of 30 inter-rater comparisons using standard guidelines.[36] Eight errors (four EQ-5D-5L, zero ICECAP-A and four ICECAP-O), eight struggles (five EQ-5D-5L, one ICECAP-A and two ICECAP-O) and 52 possible error/struggles were identified through independent rating. At the subsequent rater meeting, from the 52 possible error/struggles, a further 26 errors or struggles were agreed on: 17 of 29 for the EQ-5D-5L, 5 of 11 for the ICECAP-A and 4 of 12 for the ICECAP-O. Breakdowns of error type by measure item are reported in tables 2–4.

In total, 179 segments were generated for the EQ-5D-5L (one VAS was not completed by accident) and 150 segments each for the ICECAP-A and ICECAP-O. Twenty (11%) out of the 179 segments of the EQ-5D-5L were associated with an error and six (3%) with a struggle. Three (2%) out of the 150 segments of the ICECAP-A were associated with an error and three (2%) with a struggle. Six (4%) out of the 150 segments of the ICECAP-O were associated with an error and four (3%) with a struggle.

The majority of responses were not identified as an error or struggle on any of the three measures, indicating feasibility of use for all three PROMs. Participants found all measures easy to complete overall, showing acceptability in completing these PROMs:

> Very straightforward. (Participant 26, male, aged 65–74, dialysis patient)

> Not particularly, they all seemed, they're all pretty relevant to the questionnaire and to my condition and recovery and all that sort of thing so nothing sort of surprised me what was being asked so, happy with all the questions that was fine. (Participant 21, male, 18–35, kidney transplant outpatient)

There were more errors (17) and struggles (4) reported for EQ-5D-5L (even when excluding the EQ-VAS) than for either ICECAP measure. The most common error type for EQ-5D-5L related to judgement, with this error recorded at least once across all EQ-5D-5L dimensions. Raters decided that a judgement error had occurred when participants clearly diverged from the EQ-5D-5L instruction to focus on '*your health today*':

> I am working – I am doing this on – on a bad day. (Participant 19, male, aged 55–64, judgement errors for four of five EQ-5D-5L dimensions)

Response errors for the pain/discomfort dimension were driven by the infrequency with which they were reported to occur:

> But sitting here now I would put my state, a little bit of discomfort, but I don't think either end of the spectrum really indicates what I actually feel. Because it is a thing which either comes on and then is put right by antibiotics or painkillers, so I'm going to put moderate pain or discomfort. But perhaps there should be a box for occasional to indicate recurrent or occasional pain. (Participant 12, male, aged 65–74, pain/discomfort response error)

Response errors for usual activities were due to no response being provided and one participant felt that their true response was in between slight and moderate problems. The only other error recorded on the EQ-5D-5L was also for usual activities in terms of comprehension:

> Not sure what my usual activities are. Walking I suppose. I'm sorry I can't think what my usual activities are. So I don't know what to put there. (Participant 27, male, aged 75+, usual activities response error)

For the ICECAP-A, there were only judgement errors or struggles reported. The attachment and enjoyment dimensions were error and struggle free. Two of the judgement errors follow a similar pattern as for EQ-5D, where one individual reported their capability on a bad day, rather than at the moment. The other judgement error related to the participant's interpretation of the item:

**Table 2** Errors and struggles: EQ-5D-5L (n=30)

| | Mobility | Self-care | Usual activities | Pain/ discomfort | Anxiety/ depression | VAS* | Total |
|---|---|---|---|---|---|---|---|
| **Error** | | | | | | | |
| Comprehension | 0 | 0 | 1 | 0 | 0 | 0 | 1 |
| Retrieval | 0 | 0 | 0 | 0 | 0 | 0 | 0 |
| Judgement | 1 | 1 | 2 | 6 | 2 | 2 | 14 |
| Response | 0 | 0 | 2 | 2 | 0 | 1 | 5 |
| Struggle | 1 | 0 | 1 | 2 | 0 | 2 | 6 |
| Total | 2 | 1 | 6 | 10 | 2 | 5 | 26 |

*n=29.
VAS, Visual Analogue Scale.

… I'm reading that one as being completely independent, is that I would probably be quite happy living on my own rather than with a partner or family… (Participant 10, male, aged 65–74, judgement error)

Comprehension errors were the highest error type for ICECAP-O, with two participants unable to understand the role dimension and one participant the attachment attribute:

Question one, love and friendship, reading the supposed answers, I find them rather confusing. I can have all of the love and friendship that I want (-) not really [sure] what the question is asking. Very difficult. Totally bemused by question one, so I will hazard a guess. (Participant 8, male, aged 55–64, attachment comprehension error)

I don't want to feel valued. Again I don't understand what this means really. Valued by whom? (-) I don't know I can't answer that at all. (Participant 27, male, aged 75+, role comprehension error)

Another error on the role attribute was found when one participant focused on functioning (ie, what they do) rather than their capability (ie, what they are able to do):

Yeah, actually, it's interesting if I think about it a bit more actually. I probably am able to do all of the things that make me feel valued but don't actually do them. I think I'll leave that to many of the things.

(Participant 10, male, aged 65–74, role judgement error)

There were two response errors on the enjoyment attribute where both individuals felt they were in between the same two levels:

…don't have all the time that I'd like to spend doing stuff outside the work so my answer's probably a two and a half but I'll put a three. (Participant 16, male, aged 55–64, enjoyment response error)

Enjoyment and pleasure? I think I'm somewhere – I'm gonna put myself unhelpfully at two and a half because I don't think I have a little, I don't think I have a lot. (Participant 18, male, aged 35–44, enjoyment response error)

### Measure preferences

The EQ-5D-5L was most preferred (n=17), five preferred ICECAP-O, three preferred ICECAP-A and five were unable to make a choice. One reason for preferring EQ-5D-5L was that participants could more clearly see the connection between the questions being asked and their illness:

I think the one that…because I've come via the kidney clinic, I'm-I'm thinking that this kidney research rather than general life research, so I think the one that relates most clearly to health and different problems

**Table 3** Errors and struggles: ICECAP-A (n=30)

| | Stability | Attachment | Autonomy | Achievement | Enjoyment | Total |
|---|---|---|---|---|---|---|
| **Error** | | | | | | |
| Comprehension | 0 | 0 | 0 | 0 | 0 | 0 |
| Retrieval | 0 | 0 | 0 | 0 | 0 | 0 |
| Judgement | 0 | 0 | 2 | 1 | 0 | 3 |
| Response | 0 | 0 | 0 | 0 | 0 | 0 |
| Struggle | 2 | 0 | 0 | 1 | 0 | 3 |
| Total | 2 | 0 | 2 | 2 | 0 | 6 |

| Table 4 | Errors and struggles: ICECAP-O (n=30) | | | | | |
|---|---|---|---|---|---|---|
| | **Attachment** | **Security** | **Role** | **Enjoyment** | **Control** | **Total** |
| Error | | | | | | |
| Comprehension | 1 | 0 | 2 | 0 | 0 | 3 |
| Retrieval | 0 | 0 | 0 | 0 | 0 | 0 |
| Judgement | 0 | 0 | 1 | 0 | 0 | 1 |
| Response | 0 | 0 | 0 | 2 | 0 | 2 |
| Struggle | 1 | 1 | 2 | 0 | 0 | 4 |
| Total | 2 | 1 | 5 | 2 | 0 | 10 |

that you might experience with kidney problems… is (EQ-5D-5L). (Participant 5, female, aged 45–54)

That one about the physical thing. That seemed to be more relevant about whether you're well, ill or what other problems you've got. More relevant for a medical questionnaire rather than how you feel and stuff. But I know how you feel is important as well but you know, whether you can get about and might need help getting to appointments, things like that might be more, more relevant. (Participant 17, male, aged 35–44)

Reasons for preferring either ICECAP measure were due to what was being measured and a perceived greater depth compared with the EQ-5D-5L:

That one's (EQ-5D-5L) really quite a superficial, can I walk around, can I wash myself, kind of very operational stuff. These two are more about kind of more psychological as well as quite physical things. Other than you talk about anxiety, depression there and I instantly said I don't – clearly they're in my head I'm not depressed, I don't have that illness. I would – so would put down to these two and I would go with this one (ICECAP-A) because I quite like the – the independent, achievement and progress but I think that one was a brilliant question because I think that's probably the most important thing that is on my mind at the moment. (Participant 18, male, aged 35–44)

Because (ICECAP-O) thats… it's all embodying isn't it about your family, your life, what you do, where you think you're going. (Participant 15, male, aged 65–74)

## DISCUSSION

This study explored the response process validity, feasibility of completion and acceptability of EQ-5D-5L, ICECAP-A and ICECAP-O in patients requiring kidney care and preferences between the three PROMs. There were more errors and struggles reported with the EQ-5D-5L, mainly related to judgement errors with respect to the answer provided varying from the measure recall period '*your health today*'. Nevertheless, most participants preferred the EQ-5D-5L for reasons of ease of completion and were more directly able to link the wording of the questions to

their health condition. ICECAP-A had the fewest errors and struggles overall. One in six participants or more recorded an error or struggle in completing EQ-5D-5L pain/discomfort, EQ-5D-5L usual activities, EQ-5D-5L VAS and ICECAP-O role items.

This study is the first to collect both ICECAP-A and ICECAP-O measures simultaneously from the same population. The study benefits from having participants with a broad range of kidney problems and receiving different treatments. The heterogeneity of the sample in terms of age and treatment type means that the findings in this study could be applied to other similar settings. There are some limitations, however: the sample was predominantly male and of white ethnicity. Although most respondents were male, this is not dissimilar to renal replacement therapy recipients in the UK where almost two in every three patients (64.1%) are male.[19] Nevertheless, the findings need to be interpreted in light of the sample. In addition, the sample does not include patients receiving peritoneal dialysis or conservative care for ESKD. The think-aloud interview method also relies on participants verbalising their difficulty in task completion, so difficulties in completion that the participants did not or were not able to express are not captured here.

As with other similar size studies in different populations, this work has shown that responses to the ICECAP-A measure have fewer errors or struggles than those to EQ-5D.[27 37] It differs from the only existing comparison between EQ-5D-5L and ICECAP-O which, in a smaller study (n=10) found the EQ-5D-5L produced fewer errors in completion.[26] Errors associated with comprehending the attachment and role items on ICECAP-O are similar to previous think-aloud studies.[26 38] A potential concern over the use of the new EQ-5D-5L is the number of judgement errors that were found here. This seems to be particularly related to the prevalence of intermittent health problems for people requiring kidney care, which caused patients difficulty in responding particularly for the pain/discomfort dimension.

The findings suggest that all three measures are appropriate for use in people requiring kidney care, with low errors and struggles across all measures reflecting the feasibility and acceptability of the three PROMs in this sample. However, the three PROMs have different strengths and weaknesses; the fewer errors reported for the two ICECAP

measures may be traded against the patients' preferences for the EQ-5D-5L. Indeed, the finding that the EQ-5D-5L was preferred by patients reflects earlier works showing that patients preferred EQ-5D over a number of other condition-specific and generic measures of health status.[11] For ICECAP measures, the ICECAP-A produced fewest errors across this population requiring kidney care covering a wide range of ages, but a recent study specifically aimed at over 75 year olds requiring treatment for ESKD found ICECAP-O to be a valid outcome.[20]

From a health and care decision-making point of view, although both errors in completion and patient preferences are important in choice of measure, they are unlikely to be the only considerations for choice of measure to aid in resource allocation decisions across health and care service provision. In a recent review of EQ-5D scores (ie, using population preferences to value the relative importance of health states[7]) attached to health states for calculating QALYs in patients with ESKD, there is only a clear benefit attached to the health gain from kidney transplantation compared with other treatments, such as dialysis and conservative care.[39] This finding may not be surprising given health levels for people with kidney transplants have found to be comparable with that of the general population[40] and is generally considered the clinical 'gold standard' treatment option for people with ESKD.[41] However, what may be surprising is that the EQ-5D is not able to distinguish patient benefits from the type of dialysis, how dialysis is delivered or whether dialysis is delivered at all. Previous stated preference research from Australia has shown that pre-dialysis patients would be willing to trade-off on average 7 months of survival time to reduce the number of trips to hospital for dialysis per week and on average 15 months of survival time to reduce their restrictions on their ability to travel and make short trips.[42] Such important considerations do not appear to be captured using the current economic toolkit that focus primarily on patient health status and not the impact of that treatment on their broader ability to do and be things in life that matter to them.

Future work could look at how decision makers can use health and capability measures simultaneously in an economic evaluation. In particular for kidney care, areas where capabilities might differ most from health measures like EQ-5D could be in areas where dialysis is delivered outside of a healthcare facility (ie, peritoneal dialysis or home-based haemodialysis) or not delivered at all (ie, conservative care). This study also highlights issues surrounding the variation in interpretation and judgements relating to the framing of EQ-5D (ie, '*your health today*') and is likely to be of interest to explore further.[43]

Further research is required to better understand whether the different ICECAP measures are completed differently depending on the respondents' stage of life. Measuring capability at different stages across the life course may provide an alternative framework for using the ICECAP capability measures in economic evaluations for health and care interventions.[44] More detailed qualitative analysis of think-aloud and semistructured interviews may provide some answers in the implementation of such a life-course framework.

**Acknowledgements** We would like to thank Louise Hawkins for her help in participant recruitment and all the participants who took part in this think-aloud study. We would also like to thank Paula Lorgelly and Rachael Morton for providing an expert peer review of study protocol, Amanda Owen-Smith for taking part in a pilot think-aloud interview and the two peer reviewers for this paper. Finally, we thank Hugh McLeod for leading a discussion at the UK Health Economists' Study Group (HESG) in Bristol in June 2018 and feedback received from delegates at HESG and European Health Economics Association (EuHEA) Conference in Maastricht, the Netherlands, in July 2018.

**Contributors** PM, FC and JC developed the study design. PM, FC and JS were involved in data acquisition. All authors were involved with analysing and interpreting the data. PM initially drafted this paper. All authors (PM, FC, JS, SS, JC) were involved in the revision of the initial draft for important intellectual content and final approval of this version to be published. All authors agree to be accountable for all aspects of the work in ensuring that questions related to the accuracy or integrity of any part of the work are appropriately investigated and resolved.

**Funding** This work was initially supported by a post-doctoral fellowship co-funded by the National Institute for Health Research (NIHR) Collaboration for Leadership in Applied Health Research and Care West (NIHR CLAHRC West) and the UK Renal Registry (UKRR). The views expressed in this article are those of the authors and not necessarily those of the NHS, the NIHR, the Department of Health and Social Care or the UKRR. Time for drafting this paper for Paul Mitchell and Joanna Coast has also been supported through a Wellcome Trust investigator award (205384/Z/16/Z).

**Competing interests** JC led the development of the ICECAP-O and ICECAP-A.

**Patient and public involvement** Patients and/or the public were not involved in the design, or conduct, or reporting, or dissemination plans of this research.

**Patient consent for publication** Not required.

**Ethics approval** Ethics approval was obtained from the East of England NHS Research Ethics Committee (16/EE/0331) (see supplementary file 1 for research protocol).

**Provenance and peer review** Not commissioned; externally peer reviewed.

**Data availability statement** No data are available.

**ORCID iD**
Paul Mark Mitchell http://orcid.org/0000-0002-7593-4460

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
