## [Reviewer comments · BMJ Open]

ARTICLE DETAILS

TITLE (PROVISIONAL)	Response process validity of three patient reported outcome measures for people requiring kidney care: a think-aloud study using the EQ-5D-5L, ICECAP-A and ICECAP-O
AUTHORS	Mitchell , Paul; Caskey, Fergus; Scott, Jemima; Sanghera, Sabina; Coast, Joanna

VERSION 1 – REVIEW

REVIEWER	Melanie Hawkins Deakin University Australia
REVIEW RETURNED	15-Nov-2019

GENERAL COMMENTS	Overall: This is a well written paper with a good rationale and clear aim. The methods are well-structured and easy to follow. The results have a good range of participant quotes to support the analysis. There is a reasonable coherence of argument throughout the paper, although still requires more conclusive argument for the connection between the study aim and the results / conclusions. There is one thing I ask the authors to think about. I wonder if you are undervaluing the depth and importance of your work by calling it a study of face validity. As I understand it, face validity is not accepted by authoritative validity theorists as a form of validity evidence. In fact, it is really just a subjective opinion that a test looks right or that, on the surface, it appears to measure the appropriate content. There is no actual empirical evidence required to state that a test has face validity (https://www.americanjournalofsurgery.com/article/S0002-9610(16)30217-3/fulltext). Your research is of a much more extensive nature, and has produced evidence about the response (cognitive) processes of the interviewees as they complete the PROMs. I suggest you read about sources of validity evidence in the 2014 Standards for Educational and Psychological Testing published by the American Educational Research Association et al. and if this is difficult to get hold of then read chapters 1 and 2 of Validity and Validation in Social, Behavioural, and Health Sciences (edited by Zumbo and Chan, 2014). The Standards outline five sources of validity evidence. These are evidence based on test content, on response processes, on internal structure of an instrument, on an instrument's relations to other variables, and on the consequences of testing, as related to validity. The study you have done contributes evidence based on response processes for evaluating the extent to which the interpretation and use of scores are valid, as assessed against the constructs of the three PROMs. I recommend you read these papers:
--

	Padilla and Benítez (2014) Validity evidence based on response processes: https://www.researchgate.net/publication/259825631_VValidity_evidence_based_on_response_processes Sawatzky et al. (2017) Montreal Accord on Patient-Reported Outcomes (PROs) use: https://www.jclinepi.com/article/S0895-4356(16)30804-6/fulltext There are others (these papers will lead you to them if you are interested). I need to reveal here that I am completing a PhD on the application of validity testing theory and methodology from education and psychology to PROM validation practice. I offer these thoughts for your consideration. Some other minor points Please do a read through to pick up a few minor typos in the text such as inconsistency between the use of double and single quote marks, and with using letters or numbers when expressing numbers less than 10 (except at the start of a sentence when a word is fine). I wonder if you need to indent, italicise and use quote marks for participant quotes – it gets very busy. I think just one of these is enough. Or, if you prefer, just indent and italicise. Results: Measure preferences (p.10; line 6) Why did respondents find it difficult to choose one of the three PROMs as their most preferred? Your statement invokes curiosity in me so could you please explain why they found it difficult, or leave this sentence out and just state the preference findings. You do state the reasons for their preferences (supported by quotes) so maybe this first sentence isn't needed? Results/Discussion: (p.11; line 47) Could you please make more clear reference to 'feasibility' and 'acceptability' in the results and discussion. These words only occur when you state your study aim in the introduction and in the discussion. I see you have said that the measures are 'appropriate' for use in people requiring kidney care but how does this relate to feasibility and acceptability, and what evidence is this statement based on? You haven't directly addressed the aim in your results or discussion, even though you have nicely stated your study findings. Perhaps you could define what you mean by feasibility and acceptability, and then address how the results of the study satisfy these requirements, and why this is important.
--	---

REVIEWER	Brett Doble Duke-NUS Medical School, Singapore
REVIEW RETURNED	02-Dec-2019

GENERAL COMMENTS	1) Page 2, line 54: It would be helpful if the authors briefly described the results from the study by Shah et al (reference 20) in the Introduction and detail how the results from Shah et al. relate to their study. 2) Page 3, lines 28-35: Can the authors be more specific in why they deemed 25 patients to be sufficient for their analysis? Also, it seems that your sample under-represents patients in the 75+ group and females, what implications does this have on the interpretation of
--

	your results? Why not have a sampling approach that allowed you to capture participants from each age and sex and maybe even treatment strata equally? 3) Page 3, lines 37-39: The authors state that sampling was purposeful to achieve diversity in the type of kidney care received, but it seems that patients receiving peritoneal dialysis or conservative management without dialysis (i.e., kidney supportive care; KSC) have not been included. Is there a reason for their omission? Some discussion on how this might affect the results might also be useful. The authors also mention the inclusion of a diversity of treatments as a strength in the discussion, but it seems to me that they have not considered all possible treatments. 4) Page 3, line 60: The authors refer to the EQ-5D-3L, but provide little details as to how it is different from the 5L. Some further details might be helpful for those not familiar with the differences between the two instruments. Also given the controversy surrounding the use of the 5L in the UK the authors might want to justify why their study included the 5L as opposed to the 3L. 5) Page 4, lines 7-27: If the ICECAP-O was specifically designed for use in individuals older than 65 years why is it appropriate for the instrument to be administered to individuals younger than 65 years old and vice versa for the ICECAP-A? Is this not a kin to using the EQ-5D-3L in children or using the EQ-5D-Y in adults? Some further justification/discussion about using each instrument in populations in which it was not intended to be used should be added to the manuscript. For example, are all dimension in the ICECAP-O relevant to younger adults and vice versa for the ICECAP-A? 6) Page 6, lines 5-13: The description of the semi-structured interviews is slightly confusing. The authors specifically refer to “a part of the semi-structured interviews...” when discussion the analysis of preference between measures. Were additional questions asked during the semi-structured interviews? If so, it should be mentioned and explained why the additional results aren’t discussed in the manuscript. 7) Page 6, lines 9-13: Can the authors state that a reason for a preference between the two ICECAP measures was also requested. 8) Page 6, lines 21-27: Are the authors able to provide the characteristics of the patients who were invited or at least the 161 who responded to the follow-up? And contrast them with the 30 participants in the final sample. 9) Page 8, lines 23-26: I note that the majority of Judgement error were for the pain/discomfort dimension, all of which I assume relate to participants responding to this question based on pain they might have experienced on other days rather than the day in which they responded to the questionnaire. Given this is the highest frequency error, I suggest you clear articulate the reasons for its occurrence, specifically for the pain/discomfort dimension, rather than just providing a general example. 10) Page 11, lines 40-45: I think some additional discussion on the relatively higher frequency of Judgement error with the EQ-5D-5L should be added. What impact does this Judgement error have on the interpretation EQ-5D-5L results? How common is this type of
--	--

	error more generally in any population or specifically in patients with intermittent health problems responding to the EQ-5D and has anything been done about it? Why does the EQ-5D use such a statement whereas the other measures don't? I think this might provide a little more context for the authors' findings. 11) Page 12, lines 3-25: The authors make a lot of general and definitive statements in this section of the discussion and I agree that they are important points to consider, but some of the statements may be more applicable to certain subgroups of kidney disease patients compared to others (e.g., very elderly versus a 40 year old). Some small edits in this section might be helpful to highlight the heterogeneity in outcomes and preferences among kidney disease patients. Furthermore, my main concern with this study is that it draws broad conclusions about errors and preferences for different QoL measures from a small groups of patients for a very heterogeneous disease group (i.e., I don't think your results are necessarily generalizable to all kidney disease patients and I assume they authors don't think so either, but there is not a lot of discussion concerning this limitations or how your results would apply to different subgroups of kidney disease patients). Some discussion concerning this might be helpful.
--	--

VERSION 1 – AUTHOR RESPONSE

Reviewer: 1

Reviewer Name: Melanie Hawkins

Institution and Country:

Deakin University

Australia

Please state any competing interests or state 'None declared': None declared.

Please leave your comments for the authors below

Overall: This is a well written paper with a good rationale and clear aim. The methods are well-structured and easy to follow. The results have a good range of participant quotes to support the analysis. There is a reasonable coherence of argument throughout the paper, although still requires more conclusive argument for the connection between the study aim and the results / conclusions.

There is one thing I ask the authors to think about. I wonder if you are undervaluing the depth and importance of your work by calling it a study of face validity. As I understand it, face validity is not accepted by authoritative validity theorists as a form of validity evidence. In fact, it is really just a subjective opinion that a test looks right or that, on the surface, it appears to measure the appropriate content. There is no actual empirical evidence required to state that a test has face validity ([https://www.americanjournalofsurgery.com/article/S0002-9610\(16\)30217-3/fulltext](https://www.americanjournalofsurgery.com/article/S0002-9610(16)30217-3/fulltext)). Your research is of a much more extensive nature, and has produced evidence about the response (cognitive) processes of the interviewees as they complete the PROMs. I suggest you read about sources of validity evidence in the 2014 Standards for Educational and Psychological Testing published by the American Educational Research Association et al. and if this is difficult to get hold of then read chapters 1 and 2 of Validity and Validation in Social, Behavioural, and Health Sciences (edited by Zumbo and Chan, 2014). The Standards outline five sources of validity evidence. These are evidence based on test content, on response processes, on internal structure of an instrument, on an instrument's relations to other variables, and on the consequences of testing, as related to validity. The study you have done contributes evidence based on response processes for evaluating the extent to which the interpretation and use of scores are valid, as assessed against the constructs of the three PROMs.

I recommend you read these papers:

Padilla and Benítez (2014) Validity evidence based on response processes: https://www.researchgate.net/publication/259825631_VValidity_evidence_based_on_response_processes

Sawatzky et al. (2017) Montreal Accord on Patient-Reported Outcomes (PROs) use: [https://www.jclinepi.com/article/S0895-4356\(16\)30804-6/fulltext](https://www.jclinepi.com/article/S0895-4356(16)30804-6/fulltext)

There are others (these papers will lead you to them if you are interested).

I need to reveal here that I am completing a PhD on the application of validity testing theory and methodology from education and psychology to PROM validation practice. I offer these thoughts for your consideration.

RESPONSE 1: We thank the reviewer for this helpful suggestion. Following reading the references suggested, we agree that referring to this work as validity associated with response processes is a more accurate description of this research. We have therefore amended our title and parts of the manuscript that previously referred to “face validity” and now refers to “response process validity”. We have also added in the following sentences to our methods:

“Think aloud interviews are a method that allow for the assessment of validity in terms of investigating response processes.²⁴ Assessing response processes are one of five recommended sources of validity evidence.²⁵” (p. 3)

Some other minor points

Please do a read through to pick up a few minor typos in the text such as inconsistency between the use of double and single quote marks, and with using letters or numbers when expressing numbers less than 10 (except at the start of a sentence when a word is fine).

RESPONSE 2: Paper has been carefully re-read for typos and quotation and numbering inconsistencies. We follow the journal guidelines of spelling out numbers of less than 10 and using numbers for 10 and up.

I wonder if you need to indent, italicise and use quote marks for participant quotes – it gets very busy. I think just one of these is enough. Or, if you prefer, just indent and italicise.

RESPONSE 3: We have removed quote marks as requested.

Results: Measure preferences

(p.10; line 6) Why did respondents find it difficult to choose one of the three PROMs as their most preferred? Your statement invokes curiosity in me so could you please explain why they found it difficult, or leave this sentence out and just state the preference findings. You do state the reasons for their preferences (supported by quotes) so maybe this first sentence isn't needed?

RESPONSE 4: The reviewer is correct to say the sentence is not needed, so it has been deleted from the revision.

Results/Discussion:

(p.11; line 47) Could you please make more clear reference to ‘feasibility’ and ‘acceptability’ in the results and discussion. These words only occur when you state your study aim in the introduction and in the discussion. I see you have said that the measures are ‘appropriate’ for use in people requiring kidney care but how does this relate to feasibility and acceptability, and what evidence is this statement based on? You haven't directly addressed the aim in your results or discussion, even

though you have nicely stated your study findings. Perhaps you could define what you mean by feasibility and acceptability, and then address how the results of the study satisfy these requirements, and why this is important.

RESPONSE 5: text added to highlight feasibility and acceptability in results:

“The majority of responses were not identified as an error or struggle on any of the three measures, indicating feasibility of use for all three PROMs. Participants found all measures easy to complete overall, showing acceptability in completing these PROMs” (p. 8)

And

“The findings suggest that all three measures are appropriate for use in people requiring kidney care, with low errors and struggles across all measures reflecting the feasibility and acceptability of the three PROMs in this sample.” (p. 12)

Reviewer: 2

Reviewer Name: Brett Doble

Institution and Country: Duke-NUS Medical School, Singapore Please state any competing interests or state 'None declared': None declared

Please leave your comments for the authors below

- 1) Page 2, line 54: It would be helpful if the authors briefly described the results from the study by Shah et al (reference 20) in the Introduction and detail how the results from Shah et al. relate to their study.

RESPONSE 1: additional text added on the Shah et al. study to the introduction and discussion:

“A recent study found the ICECAP-O to be a valid measure in over 75 year old patients receiving dialysis or conservative care for end stage kidney disease (ESKD).²⁰” (p. 2)

“For ICECAP measures, the ICECAP-A produced fewest errors across this population requiring kidney care covering a wide range of ages, but a recent study specifically aimed at over 75 year olds requiring treatment for ESKD found ICECAP-O to be a valid outcome.²⁰” (p. 12)

2) Page 3, lines 28-35: Can the authors be more specific in why they deemed 25 patients to be sufficient for their analysis? Also, it seems that your sample under-represents patients in the 75+ group and females, what implications does this have on the interpretation of your results? Why not have a sampling approach that allowed you to capture participants from each age and sex and maybe even treatment strata equally?

RESPONSE 2: Our sampling and recruitment section has been revised to address the reviewers' comments. We believed that, based on previous research, saturation would be reached at approximately 25 participants:

“Samples for previous think-aloud studies on health and capability PROMs have ranged from 10²⁶ to 34²⁷ participants. Based on these previous studies, saturation (whereby no new insights would be anticipated from additional sampling)²⁸ was expected to be reached at 25 participants here.” (p. 3)

Sampling for qualitative research studies often attempts to be purposeful, rather than a representative sample that you would typically find in an intervention study:

“Participants were sampled purposefully to achieve diversity in age (classified as <65 or >65) and type of kidney care received, but in line with general approaches to sampling in qualitative research sampling, did not aim for representativeness.”²⁸ (p. 3)

3) Page 3, lines 37-39: The authors state that sampling was purposeful to achieve diversity in the type of kidney care received, but it seems that patients receiving peritoneal dialysis or conservative management without dialysis (i.e., kidney supportive care; KSC) have not been included. Is there a reason for their omission? Some discussion on how this might affect the results might also be useful. The authors also mention the inclusion of a diversity of treatments as a strength in the discussion, but it seems to me that they have not considered all possible treatments.

RESPONSE 3: The following text has been added to the discussion to recognise the limitation raised by the reviewer that not all patients receiving kidney care were included in this study:

“In addition, the sample does not include patients receiving peritoneal dialysis or conservative care for ESKD.” (p. 12)

AND

“Future work could look at how decision-makers can use health and capability measures simultaneously in an economic evaluation. In particular for kidney care, areas where capabilities might differ most from health measures like EQ-5D could be in areas where dialysis is delivered outside of a health care facility (i.e. peritoneal dialysis or home-based haemodialysis) or not delivered at all (i.e. conservative care).” (p. 13)

4) Page 3, line 60: The authors refer to the EQ-5D-3L, but provide little details as to how it is different from the 5L. Some further details might be helpful for those not familiar with the differences between the two instruments. Also given the controversy surrounding the use of the 5L in the UK the authors might want to justify why their study included the 5L as opposed to the 3L.

RESPONSE 4: Additional text has been added to emphasise the main differences in measures:

“The EQ-5D-5L was introduced to supersede the EQ-5D-3L to reduce ceiling effects and increase sensitivity to change, by moving from a three level to a five level severity measure of health problems, but with the same five dimensions.” (p. 4)

Although there has been controversy around the valuation of the EQ-5D-5L measure for estimating quality adjusted life years for economic evaluation, NICE still recommends collection of the EQ-5D-5L descriptive system, and so we have added text to highlight this:

“The National Institute for Health and Care Excellence (NICE) in England recommends the collection of the EQ-5D-5L for conducting economic evaluation in clinical studies.³⁰” (p. 4)

We think it is beyond the scope of this study to discuss the issues with the valuation of EQ-5D-5L any further here, although we have mentioned limitations of the descriptive system as a point of consideration in future research:

“This study also highlights issues surrounding the variation in interpretation and judgements relating to the framing of EQ-5D (i.e. “*your health today*”) and is likely to be of interest to explore further.⁴³” (p.13)

5) Page 4, lines 7-27: If the ICECAP-O was specifically designed for use in individuals older than 65 years why is it appropriate for the instrument to be administered to individuals younger than 65 years old and vice versa for the ICECAP-A? Is this not a kin to using the EQ-5D-3L in children or using the EQ-5D-Y in adults? Some further justification/discussion about using each instrument in populations in which it was not intended to be used should be added to the manuscript. For example, are all dimension in the ICECAP-O relevant to younger adults and vice versa for the ICECAP-A?

RESPONSE 5: the ICECAP-A is developed for all adults, not just those aged between 18-65. We have added extra text to make this clearer for the reader:

“The ICECAP-A is a capability wellbeing measure developed for the general adult population (i.e. all adults, including those aged over 65). It consists of five dimensions relating to a person’s capability to have attachment, stability, achievement, enjoyment and autonomy. Each dimension has four levels ranging from no capability to full capability. The capabilities were identified through qualitative research with members of the general public aged 18 years and above (including over 65 year olds) to identify what was most important to them in their life.” (p. 4)

We have also added further justification for looking at both A and O in this sample to the introduction:

“ICECAP-O was also developed first and has been shown to be a valid outcome in older and younger adults in different settings.” (p. 2)

6) Page 6, lines 5-13: The description of the semi-structured interviews is slightly confusing. The authors specifically refer to “a part of the semi-structured interviews...” when discussion the analysis of preference between measures. Were additional questions asked during the semi-structured interviews? If so, it should be mentioned and explained why the additional results aren’t discussed in the manuscript.

RESPONSE 6: We do agree that the sentence could have been structured more clearly and have amended the sentence to alleviate confusion for other readers:

“During the semi-structured interviews following the think-aloud task, individual preferences for completing the three measures were explored.” (p. 6)

What was included in the semi-structured interviews was explained previously in the manuscript:

“Following the think-aloud task, a semi-structured interview was conducted to clarify issues arising in the think-aloud task and to explore views about the three measures.” (p. 5)

7) Page 6, lines 9-13: Can the authors state that a reason for a preference between the two ICECAP measures was also requested.

RESPONSE 7: We have removed this sentence in response to a comment from reviewer 1.

8) Page 6, lines 21-27: Are the authors able to provide the characteristics of the patients who were invited or at least the 161 who responded to the follow-up? And contrast them with the 30 participants in the final sample.

RESPONSE 8: We can only provide data on patients who consented to take part in this study and so cannot compare with those who did not agree to participate in the study. We do, however, try to compare this sample with the prevalent UK renal replacement therapy population in our discussion:

“There are some limitations, however: the sample was predominantly male and of white ethnicity. Although most respondents were male, this is not dissimilar to renal replacement therapy recipients in the UK where almost two in every three patients (64.1%) are male.¹⁹” (p.12)

9) Page 8, lines 23-26: I note that the majority of Judgement error were for the pain/discomfort dimension, all of which I assume relate to participants responding to this question based on pain they might have experienced on other days rather than the day in which they responded to the questionnaire. Given this is the highest frequency error, I suggest you clearly articulate the reasons for its occurrence, specifically for the pain/discomfort dimension, rather than just providing a general example.

RESPONSE 9: The example provided in the paper is a typical example of a judgement error both for pain/discomfort and the other EQ-5D dimensions. We think it is a clear demonstration of people not reporting EQ-5D dimensions for “today” – for example “on a bad day” - and so have kept this quotation. We have added text on the frequency of errors for pain/discomfort to the discussion section:

“A potential concern over the use of the new EQ-5D-5L is the number of judgement errors that were found here. This seems to be related to the prevalence of intermittent health problems for people requiring kidney care, which caused patients difficulty in responding particularly for the pain/discomfort dimension.” (p. 12)

10) Page 11, lines 40-45: I think some additional discussion on the relatively higher frequency of Judgement error with the EQ-5D-5L should be added. What impact does this Judgement error have on the interpretation of EQ-5D-5L results? How common is this type of error more generally in any population or specifically in patients with intermittent health problems responding to the EQ-5D and has anything been done about it? Why does the EQ-5D use such a statement whereas the other measures don't? I think this might provide a little more context for the authors' findings.

RESPONSE 10: We agree with the reviewer that there are important points to be addressed further, but we think it is beyond the scope of our study to give definite answers to all of the points the reviewer has raised. We suggest this as an area to explore further in future research.

“This study highlights issues surrounding the variation in interpretation and judgements relating to the framing of EQ-5D (i.e. “*your health today*”) and is likely to be of interest to explore further.⁴³” (p.13)

11) Page 12, lines 3-25: The authors make a lot of general and definitive statements in this section of the discussion and I agree that they are important points to consider, but some of the statements may be more applicable to certain subgroups of kidney disease patients compared to others (e.g., very elderly versus a 40 year old). Some small edits in this section might be helpful to highlight the heterogeneity in outcomes and preferences among kidney disease patients. Furthermore, my main concern with this study is that it draws broad conclusions about errors and preferences for different

QoL measures from a small groups of patients for a very heterogeneous disease group (i.e., I don't think your results are necessarily generalizable to all kidney disease patients and I assume they authors don't think so either, but there is not a lot of discussion concerning this limitations or how your results would apply to different subgroups of kidney disease patients). Some discussion concerning this might be helpful.

RESPONSE 11: In response to this comment as well as the Editor's recommendation, we have explicitly noted that a limitation of this study is that it is unable to cover all kidney patient populations:

"In addition, the sample does not include patients receiving peritoneal dialysis or conservative care for ESKD." (p. 12)

We do agree that the findings may not be generalizable to all kidney disease patients and this was not the aim of this study. We do think, however, that the typicality of our findings may be applicable to other patient groups that have a heterogeneous sample like ours and so have added the following text to demonstrate this more clearly:

"The heterogeneity of the sample in terms of age and treatment type means that the findings in this study could be applied to other similar settings." (p. 12)

VERSION 2 – REVIEW

REVIEWER	Melanie Hawkins Deakin University, Australia
REVIEW RETURNED	10-Feb-2020
GENERAL COMMENTS	I'm glad you found the suggestion about evidence based on response processes to be useful. I feel it better reflects the depth of investigation you have undertaken. Your edits have improved the paper and made the connection between aim and results more clear. I am recommending your paper be accepted for publication.